# Complete Genome Sequence and Benzophenone-3 Mineralisation Potential of *Rhodococcus* sp. USK10, A Bacterium Isolated from Riverbank Sediment

**Joseph Donald Martin** [1], **Urse Scheel Krüger** [2], **Athanasios Zervas** [1], **Morten Dencker Schostag** [2,3], **Tue Kjærgaard Nielsen** [4], **Jens Aamand** [2], **Lars Hestbjerg Hansen** [4] and **Lea Ellegaard-Jensen** [1,*]

[1] Department of Environmental Science, Faculty of Technical Science, Aarhus University, 4000 Roskilde, Denmark; jdmar@envs.au.dk (J.D.M.); az@envs.au.dk (A.Z.)

[2] Department of Geochemistry, Geological Survey of Denmark and Greenland (GEUS), 1350 Copenhagen, Denmark; usk@geus.dk (U.S.K.); mdesc@dtu.dk (M.D.S.); jeaa@geus.dk (J.A.)

[3] Department of Biotechnology and Biomedicine, Technical University of Denmark, 2800 Kongens Lyngby, Denmark

[4] Department of Plant and Environmental Sciences, University of Copenhagen, 1870 Frederiksber, Denmark; tkn@plen.ku.dk (T.K.N.); lhha@plen.ku.dk (L.H.H.)

* Correspondence: leael@envs.au.dk

**Abstract:** Benzophenone-3 (BP3) is an organic UV filter whose presence in the aquatic environment has been linked to detrimental developmental impacts in aquatic organisms such as coral and fish. The genus *Rhodococcus* has been extensively studied and is known for possessing large genomes housing genes for biodegradation of a wide range of compounds, including aromatic carbons. Here, we present the genome sequence of *Rhodococcus* sp. USK10, which was isolated from Chinese riverbank sediment and is capable of utilising BP3 as the sole carbon source, resulting in full BP3 mineralisation. The genome consisted of 9,870,030 bp in 3 replicons, a G+C content of 67.2%, and 9722 coding DNA sequences (CDSs). Annotation of the genome revealed that 179 of these CDSs are involved in the metabolism of aromatic carbons. The complete genome of *Rhodococcus* sp. USK10 is the first complete, annotated genome sequence of a Benzophenone-3-degrading bacterium. Through radiolabelling, it is also the first bacterium proven to mineralise Benzophenone-3. Due to the widespread environmental prevalence of Benzophenone-3, coupled with its adverse impact on aquatic organisms, this characterisation provides an integral first step in better understanding the environmentally relevant degradation pathway of the commonly used UV filter. Given USK10's ability to completely mineralise Benzophenone-3, it could prove to be a suitable candidate for bioremediation application.

**Keywords:** oxybenzone; UV filter; biodegradation; whole-genome sequencing; *Rhodococcus*

## 1. Introduction

Benzophenone-3 (BP3; 2-hydroxy-4-methoxybenzophenone; Oxybenzone) is an organic UV filter typically used in personal care products to protect the skin from harmful solar radiation. Organic UV filters have an aromatic chemical structure that allows for the absorption and stabilisation of both UVA (315–400 nm) and UVB (280–315 nm) radiation [1]. BP3 has been implemented as an active ingredient in sunscreens, cosmetics, and plastic products for decades and is still one of the most commonly used UV filters worldwide. BP3 has been detected in surface waters, sediments, and organisms within various environments, including remote areas such as seawater of the Polar Regions [2,3]. Elevated concentrations of BP3 in the aquatic environment have been reported to result in adverse effects on aquatic organisms, such as deterioration of coral reefs and impaired reproduction potential in fish [1,4–6]. These detrimental factors have caused the use of BP3-containing sunscreens to be banned on the coasts of several countries, including the United States

(Virgin Islands, HI, USA), Mexico, and Palau [1,4,7]. The chemical characteristics of BP3, and many other organic UV filters, is a cause of concern due to their high lipophilicity, allowing for them to easily bioaccumulate in aquatic organisms and even in the body fluids of humans [2,8]. In addition, BP3 may also act as an endocrine disruptor in humans, influencing birth weight and gestational age [9]. The presence of BP3 in the aquatic environment worldwide begs the question of its persistence, and therefore, it is important to further research the biodegradation potential of BP3 facilitated by microorganisms found in natural environments.

In this study, we isolated and characterised the genome of *Rhodococcus* sp. USK10, to provide additional evidence of the genetic background of this BP3-mineralising bacterium. Currently, only two other bacterial strains, *Methylophilus* sp. Strain FP-6 [10] and *Sphingomonas wittichii* strain BP14P [11], have been reported capable of degrading BP3. The phylogenetic characterisation of these strains was, however, solely based on 16S rRNA gene sequences, and their genetic makeup was not investigated. Furthermore, both strains were hypothesised to be able to mineralise BP3, without, however, confirming it experimentally.

Here, we present the first complete and annotated genome of a BP3 degrader found in nature, including a potential linear megaplasmid and a smaller circular plasmid. Strain USK10 shows an increased number of genes involved in catalysing aromatic compounds compared to related *Rhodococcus* strains, which may indicate that it is a specialist strain. In addition, we present experimental data that prove the biodegradation of BP3 by *Rhodococcus* sp. USK10, when incubated in liquid media without any other carbon source.

## 2. Materials and Methods

### 2.1. Isolation of Rhodococcus sp. USK10

Strain USK10 was isolated from enrichment cultures originating from Chinese riverbank sediment (GPS coordinates: 25.569611, 119.781000). In short, the sediment was implemented into a series of enrichment cultures using radiolabelled BP3 to assess degradation potential, followed by a series of streak plating using BP3-enriched agar plates as the sole carbon source. Single colonies were picked and further assessed for BP3 mineralisation potential and later characterised, one of which being strain USK10.

### 2.2. BP3 Biodegradation Experiment

Precultures for the mineralisation experiment were grown on R2B media supplemented with 100 ppm BP3. After incubation at 20 °C in the dark on an orbital shaker (120 rpm) for 3 days, extracts were centrifuged (12,000 × $g$, 5 min), washed twice, and resuspended in Difco$^{TM}$ Bushnell–Haas Broth (BHB). The mineralisation experiment was conducted in triplicate with each microcosm containing 5 mL of BHB with BP3 as the sole carbon source. Each USK10 microcosm had approximately $1.4 \times 10^8$ cells, while the abiotic control had no cells. The initial BP3 concentration of each microcosm was 10 mg L$^{-1}$, including [benzene-$^{14}$C(U)]-labelled BP3 (Moravek Biochemicals Inc.; Brea, CA, USA) amounting to 7055 DPM. The flasks further contained a 2 mL glass tube with 1 mL 1M NaOH serving as a basetrap to capture the evolved $^{14}$CO$_2$ during BP3 mineralisation. The microcosms were incubated in the dark at 20 °C and sampled once a day for 10 days. At each sampling time point, the NaOH was removed, replaced, and transferred to a plastic scintillation vial containing 10 mL of OptPhase HiSafe 3 scintillation cocktail (PerkinElmer, Waltman, MA, USA). All vials were counted for 10 min using a Tri-Carb 2810 TR liquid scintillation analyser (PerkinElmer, Waltman, MA, USA).

### 2.3. DNA Extraction and Library Preparation

High-molecular-weight DNA was extracted from USK10 grown on R2B liquid media. Prior to DNA extraction, strain purity was confirmed via streak plating on agar plates containing BP3 at a concentration of 250 ppm. DNA extractions were conducted using the Genomic Mini AX Bacteria kit (A&A Biotechnology, Gdynia, Poland). After extraction, the DNA was cleaned and concentrated using the Genomic DNA Clean & Concentrator kit

(Zymo Research, Irvine, CA, USA) to remove any impurities that may have been present in the extracts. Concentration and quality of the DNA extracts were measured using a Qubit 2.0 Fluorometer with the 1x DS DNA Assay (Invitrogen, Carlsbad, CA, USA) and NanoDrop Spectrophotometer ND-1000 (Thermo Fisher Scientific, Walther, MA, USA), respectively. For Illumina sequencing, an Illumina Nextera XT library was prepared for paired-end sequencing on an Illumina NextSeq550 (Illumina Inc., San Diego, CA, USA) according to the manufacturer's protocol. For Oxford Nanopore sequencing, a library was prepared using the Rapid Sequencing kit (SQK-RBK004) according to the manufacturer's instructions. Sequencing was performed on a MinION (Oxford Nanopore Technologies, Oxford, UK) with a FLO-MIN106 flow cell, controlled using MinKNOW (19.10.1).

### 2.4. Bioinformatics Analyses

Sequencing adapters for Illumina reads were trimmed with Trim Galore (0.6.4) (https: //github.com/FelixKrueger/TrimGalore (accessed on 5 March 2020)). Raw Nanopore fast5 reads were basecalled with GPU-Guppy (3.2.6+afc8e14). A long-read-only assembly was created using Raven (1.2.2) [12] and subsequently polished with the Unicycler polish module from the Unicycler assembler (0.4.8) [13], which applies long-read polishing with Racon [14] and short-read polishing with Pilon [15]. The completeness of the genomes was verified by mapping to reference using the Illumina and Nanopore reads with BBmap [16] and Minimap2 [17] under default settings, implemented in Geneious Prime v2020.2.4 (Biomatters). Plasmid sequences were classified using MOB-suite (3.0.0) [18]. The assembled genome was annotated using Rapid Annotation using Subsystem Technology (RAST), an online prokaryotic genome annotation platform [19]. Genome completeness was evaluated using BUSCO v5.2.2 using the bacteria_odb10 lineage and "genome" mode [20]. For 16S phylogenetic tree construction, 16S rRNA gene sequences of strains related to USK10 were retrieved by BLAST (https://blast.ncbi.nlm.nih.gov/Blast.cgi (accessed on 26 May 2021)). The 16S rRNA gene sequences were collected and aligned with MAAFT [21] under default settings in Geneious Prime v2020.2.4. The alignment was subsequently used for the construction of a 16S rRNA-based phylogenetic tree using RAxML [22] in Geneious Prime, specifically the Rapid Bootstrapping and search for best-scoring ML tree algorithm with 100 iterations. Whole genome-based phylogenetic analysis was conducted using the Genome Taxonomy Database (GTDB) [23,24]. The classify workflow (classify_wf) of the Genome Taxonomy Database Toolkit (GTDB-Tk) [25] was used to determine USK10's taxonomic assignment. The workflow produced a list of genomes similar to that of USK10 along with ANI scores for comparison purposes. Those genome assemblies were retrieved via NCBI and implemented in the lineage workflow (lineage_wf) of CheckM to assess the similarities of their core genomes [26]. The alignment produced via CheckM was uploaded to Geneious Prime. A phylogenetic tree was created using RAxML, which utilised the GTR GAMMA nucleotide substitution model under the "Rapid Bootstrapping and search for besting-scoring ML tree" algorithm with 100 bootstraps replicated.

### 3. Results and Discussion

#### 3.1. BP3 Degradation Potential of Rhodococcus sp. USK10

BP3 mineralisation potential of *Rhodococcus* sp. USK10 was evaluated by measuring released carbon dioxide originating from labelled BP3 added as a sole carbon source in a liquid medium microcosm. Figure 1 depicts the complete mineralisation of BP3 by strain USK10. USK10 starts to mineralise BP3 following a two-day lag phase. On the 10th day of the experiment, cumulatively 52.7% of the initial 14C label was collected in the form of $^{14}CO_2$. The remaining labelled carbon fraction has likely been incorporated into the construction of cellular biomass or metabolites [27]. Comparatively, Lui and colleagues [28] studied biodegradation of BP3 in activated sludge microcosms, focusing on the biodegradation under various redox conditions. They reported that BP3 was completely biodegraded within 42 days of incubation. However, the half-lives of BP3 were observed to be relatively shorter at approximately 4–11 days. In the present study, *Rhodococcus* sp. USK10 demon-

strates the ability to mineralise BP3 within 10 days under aerobic conditions. Alternatively, degradation of BP3 has been shown in water via the UV/$H_2O_2$ and UV/persulfate (UV/PS) reactions, but also using persulfate, metal ions, $PbO/TiO_2$ and $Sb_2O_3/TiO_2$, and other chemicals [29–31]. However, these solutions are not considered "green solutions".

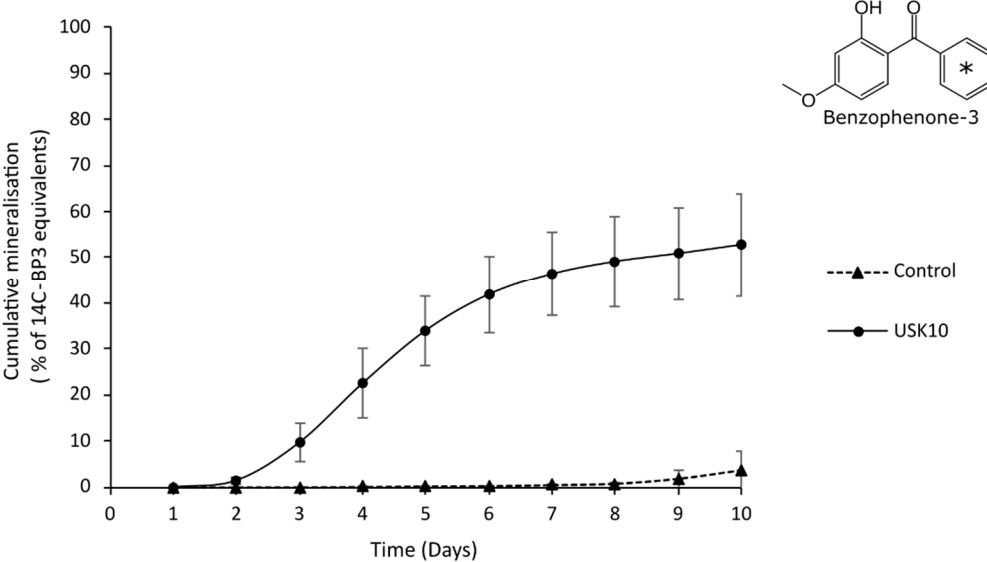

**Figure 1.** Cumulative mineralisation of BP3 by strain USK10 in pure culture and an abiotic control over ten days. Mean values and standard deviation based on three replicates are shown for $^{14}CO_2$ production relative to the initial amount of $^{14}C$-BP3 added ($^{14}C0$). The structure of BP-3 is illustrated on the top right; the $^{14}C$-labelling is indicated by a *.

In order to pursue a potential green alternative to chemical water treatment of BP3, the authors suggest that future research should focus on the gained benefits of applying Rhodococcus sp. USK10 in bioaugmentation campaigns compared to the natural attenuation of BP3 in a given environment, since this is presently lacking in the literature. In comparison, Kundu et al. [32] described the bioremediation potential of *Rhodococcus pyridinivorans* NT2 in soils contaminated by nitrotoluene. They found that a combination of bioaugmentation and biostimulation significantly increased biodegradation compared to natural attenuation.

Furthermore, due to its genetic composition, USK10 may also be capable of degrading other aromatic compounds, as has been shown for other *Rhodococcus* strains [33], and could potentially be used for bioremediation of other organic pollutants as well. However, as pointed out by Martinkova et al. [33], mere genetic diversity of catabolic genes and pathways may not be directly transferable to in situ bioremediation abilities. Future studies are needed to confirm the full catabolic range of *Rhodococcus* sp. USK10.

### 3.2. Genome Analysis

The complete genome sequence of *Rhodococcus* sp. USK10 is composed of three replicons with a total assembly of 9,870,030 bp and a G+C content of 67.2%. The chromosome is 8,396,788 bp (G+C content: 67.6%), while the two mobilisable plasmids are 1,355,759 bp (linear, G+C content: 64.6%) and 117,483 (circular, GC content: 66%). The genomic map of the chromosome and the two plasmids are presented in Figure 2. The circularity of the three replicons was verified by mapping-to-reference runs using the Illumina and Nanopore reads in Geneious Prime. For the chromosome and the small plasmid, these were successful. For the larger plasmid, manual forcing of circularity in Geneious Prime and subsequent mapping-to-reference yielded negative results for both the Illumina and Nanopore reads. *Rhodococcus* spp., as well as other Actinobacteria (e.g., *Micrococcus* spp. [34]), are known for having large linear plasmids housing genes coding for

degradation potential [33,35,36]. The assembly is of high quality, as revealed by BUSCO analysis (123 complete BUSCOs/120 single-copy and 3 double BUSCOs/1 fragmented BUSCO/99.2% coverage of bacteria_odb10).

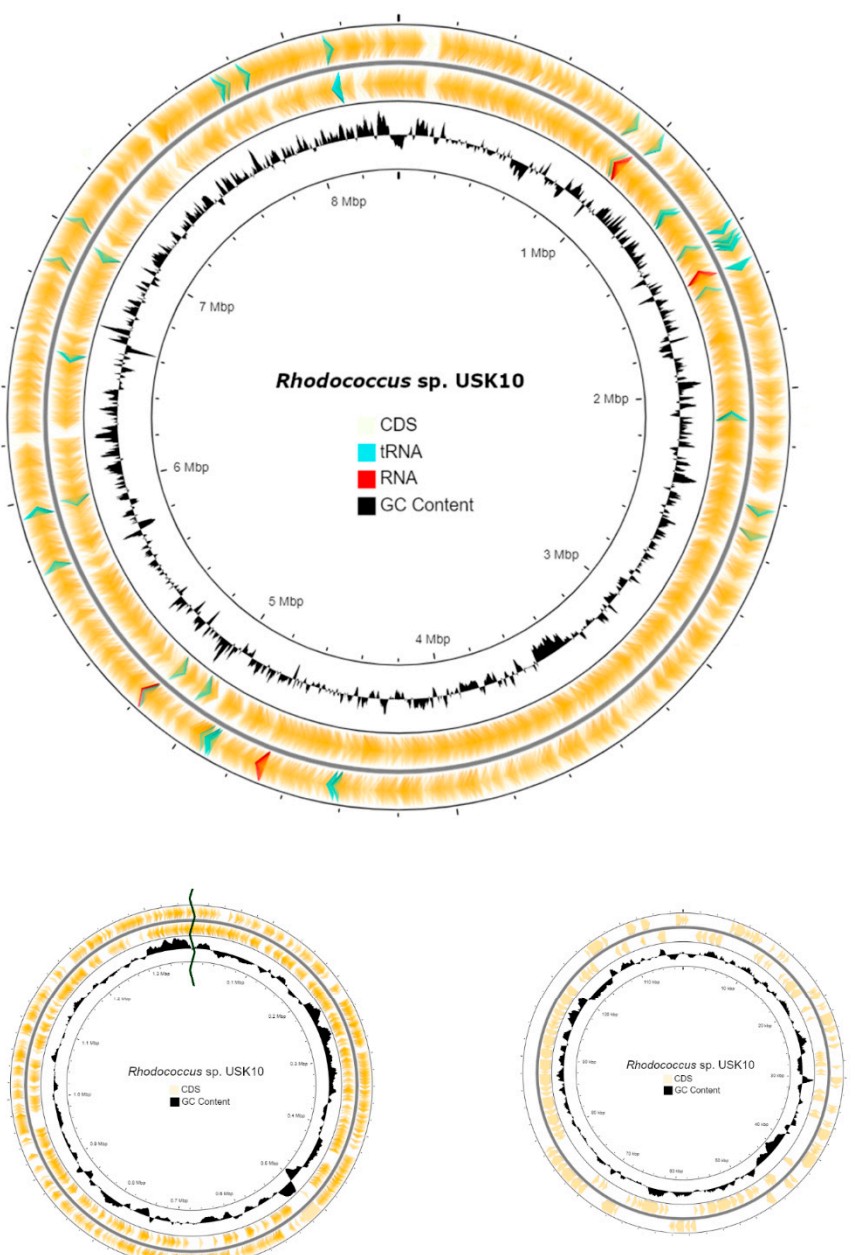

**Figure 2.** Circular map of *Rhodococcus* sp. USK10 chromosome. The two outer rings represent the coding sequences of the chromosome; the outermost being the forward strand and the innermost being the reverse strand. The inner most ring represents GC content. The G+C content of the chromosome is 67.2%. Created using CGview Server [37].

### 3.3. Phylogenetic Placement of Rhodococcus sp. USK10

The phylogenetic analysis of both the 16S rRNA gene sequences and whole genome showed that USK10 is well supported within the *Rhodococcus* genus. Based on 16s rRNA gene sequences, *R. wratislaviensis* DSM 44,107 and *R. koreensis* DNP505 are the closest relatives of USK10, having pairwise identities of 99.5% and 99.2%, respectively. Figure 3 presents the phylogenetic tree based on 16S rRNA gene sequences.

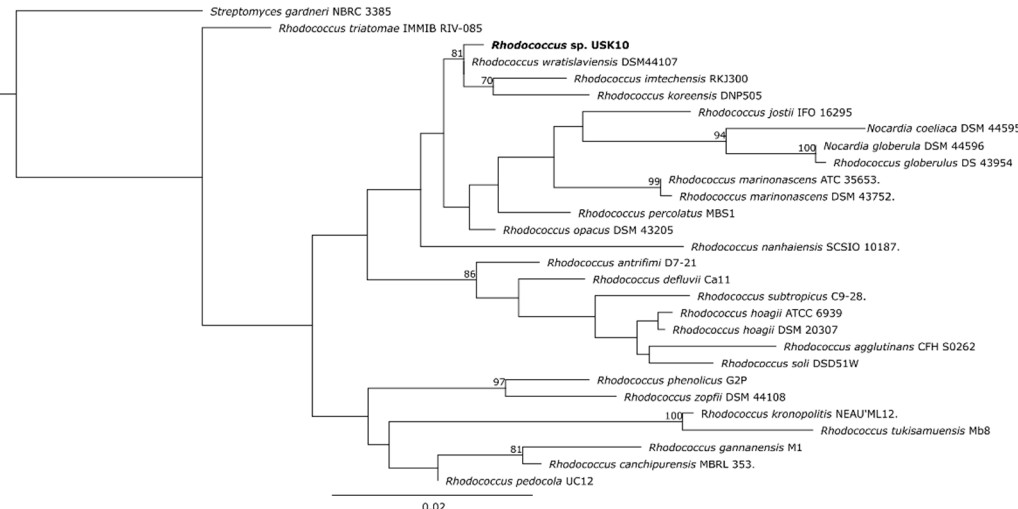

**Figure 3.** A phylogenetic tree based on 16S rRNA gene sequences showing the position of USK10 in relation to other *Rhodococcus* species and related genera of Actinobacteria. The nucleotide sequences were obtained from NCBI and aligned using the MAAFT alignment plugin via Geneious Prime 2020.2.4. The tree was constructed by RAxML (version 8.2.11) in Geneious Prime 2020.2.4. Node numbers denote bootstrap support values above 60%. The nucleotide substitution mode used was GTR GAMMA, and the algorithm used was "Rapid Bootstrapping and search for the best-scoring ML tree" with 100 replicates. The distance scale indicates 0.02 substitutions per nucleotide position.

For whole-genome analysis, GTDB-Tk classified the bacterial genomes based on phylogeny of 120 marker genes and ANI [25]. The ANI scores of each genome in relation to USK10 are presented jointly with the CheckM whole-genome tree in Figure 4. From this analysis, the *Rhodococcus* strain determined to be the most related to USK10 was *Rhodococcus* sp. NCIMB 12,038 with an ANI score of 95. This borders the species' demarcation threshold [38]. The second-most related bacterium was *Rhodococcus koreensis* DSM 44489, which had an ANI score of 94.92. Considering the limited number of available *Rhodococcus* genomes, the exact ANI threshold for species affiliation is not certain. It has been seen on other bacterial groups (e.g., the Bacillus cereus group [39], the genus Serratia [40]) that this threshold ranges between 92 and 96%. Based on the topology of both the 16S rRNA sequences and the whole-genome comparison, USK10 can be definitely placed and is well supported within the *Rhodococcus* genus. Additional characterisation analyses, such as chemotaxonomic and biochemical assays, which are outside the scope of this study, would need to be conducted for a confident taxonomic assignment of strain USK10, as well as strain NCIMB 12038.

### 3.4. Genome Annotation

The annotated genome contains a total of 9722 CDSs along with 61 RNA-encoding genes. RAST was able to provide a general overview of the biological features within the genome, achieving a subsystem coverage of 39% of the annotated genes, corresponding to 3817 subsystem feature counts. Of those counts, 179 were responsible for metabolism of aromatic compounds, some of which are likely involved in the degradation process of BP3. Four classes of enzymes involved in the metabolism of aromatic compounds (26 hydrolases, 5 isomerases, 15 lyases, and 63 oxidoreductases,) were annotated in the coding sequences. Additionally, 13 transfer proteins involved in degradation processes were identified. Of the remainder of the CDSs annotated for involvement in metabolism of aromatic compounds, two were part of the PcA regulatory protein PcAR family and 39 part of the transcriptional regulator IclR family. Both these protein families have been well documented to be involved in the degradation of aromatic hydrocarbons in other *Rhodococcus* species [41]. In *Sphingomonas wittichii* RW1 and DC-6, the first step in the degradation of aromatic compounds is performed by a dioxygenase gene located on a megaplasmid [42,43]. USK10

bears six dioxygenases on its linear megaplasmid, one on the small circular plasmid, and 65 dioxygenases on its chromosome. Interestingly, the dioxygenase on the small plasmid (3-carboxyethylcatechol 2,3-dioxygenase) is placed next to a FAD-binding monooxygenase (PheA/TfdB family, similar to 2,4-dichlorophenol 6-monooxygenase), which is involved in the degradation of another phenolic compound, 2,4-dichlorophenol. As another alternative, hydroxylases have been suggested to be implemented in the first step of BP3 biodegradation [30]. USK10 possesses 11 hydroxylases on its megaplasmid, one on its small plasmid, and 26 on its genome. The megaplasmid contains 65 oxidoreductases that may also play a role in USK10′s biodegradation potential of BP3. Further exploitation of the *Rhodococcus* sp. USK10 genome, and that of other degraders, could lead to more confident identification of potential genes and processes involved in the biodegradation of BP3. Transcriptome sequencing and potentially proteomics analysis of BP3 degrading bacteria may also illuminate the involved genes, if expression of the involved catabolic genes is regulated during BP3 degradation. So, with this characterisation of *Rhodococcus* sp. USK10, we have provided an integral first step in understanding the environmentally relevant biodegradation of the commonly used UV filter BP3.

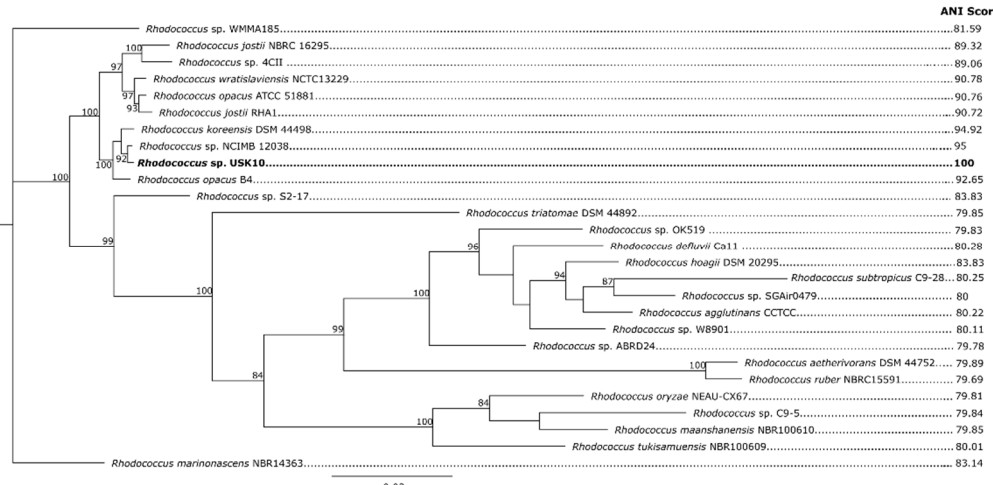

**Figure 4.** Phylogenetic tree constructed around the position of USK10 based on whole-genome sequences using CheckM alignment. The genome sequences were obtained via the NCBI assembly database. The tree was constructed by RAxML (version 8.2.11) in Geneious Prime 2.0. The nucleotide substitution model used was GTR GAMMA, and the algorithm used was "Rapid Bootstrapping and search for the best-scoring ML tree" with 100 replicates. Node numbers denote bootstrap support values above 80%. Score values on the right indicate ANI scores obtained via whole-genome comparison of USK10 using the GTDB-Tk classify workflow. The distance scale indicates 0.03 substitutions per nucleotide position.

**Author Contributions:** Conceptualisation: J.D.M. and U.S.K.; Methodology: J.D.M., A.Z., U.S.K. and T.K.N.; Validation: J.D.M. and U.S.K.; Resources: J.A., L.H.H. and L.E.-J.; Writing: J.D.M., A.Z., U.S.K. and T.K.N.; Review and Editing: J.D.M., A.Z., U.S.K., M.D.S., T.K.N., J.A., L.H.H. and L.E.-J.; Supervision: L.E.-J., M.D.S., J.A. and L.H.H. All authors have read and agreed to the published version of the manuscript.

**Funding:** This project was funded by Aarhus University Research Foundation starting grant (AUFF-E-2017-7-21) and Rural Water and Food Security (PI RURAL), the European Commission (Contract No. PI/2017/382/-112).

**Institutional Review Board Statement:** Not applicable.

**Informed Consent Statement:** Not applicable.

**Data Availability Statement:** The genome and plasmid sequence of *Rhodococcus* sp. USK10 has been deposited in GenBank under Accession Numbers CP076046-CP076048. The accession numbers of related *Rhodococcus* genomes used in this study, downloaded from NCBI, are: GCF_900105905.1; GCF_000010805.1; GCF_900455735.1; GCF_012396235.1; GCF_000014565.1; GCF_001894825.1; GCF_014 256275.1; GCF_003130705.1; GCF_001894885.1; GCF_001767395.1; GCF_000738775.1; GCF_005434945.1; GCF_004011865.1; GCF_013348805.1; GCF_001646645.1; GCF_001894985.1; GCF_005484805.1; GCF_011 058165.1; GCF_014217785.1; GCF_001894865.1; GCF_006704125.1; GCF_003051005.1; GCF_005049235.1; GCF_004328705.1; GCF_001894945.1.

**Conflicts of Interest:** No conflict of interest are declared.

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
