# Peer review of "Complete Genome Sequence and Benzophenone-3 Mineralisation Potential of Rhodococcus sp. USK10, A Bacterium Isolated from Riverbank Sediment"

_2673-8007, doi:10.3390/applmicrobiol2010006_

Round 1

Reviewer 1 Report

The work by Martin et al described the isolation of benzophenone-3 (BP3) degrading strain Rhodococcus sp. USK10 and the identification by 16S rRNA and ANI analysis. This is a nice paper in the topic. Here are some minor comments.

  1. A chemical structure of BP3 was recommended to be shown in Fig 1, since not all readers realize what benzophenone-3 is.
  2. 2. two plasmids should be shown. They were long enough..
  3. 3 and 4. The public database accession No.s should be shown in order to make readers to repeat them.
  4. Some references were cited incorrectly. For example, line 229-231, original refs for two Sphingomonas strains should be cited.
  5. Line 261, do they have names of two plasmids? Give the name and corresponding accession numbers.
  6. References should be in the same style. For example, Journal names of ref 4 and 34 were in short form
  7. Is a CONCLUSION section needed?

Author Response

We thank reviewer 1 for their valuable input. We have gone meticulously through the each comment, and corrected the manuscript accordingly. Please see the detailed replies to each comment below.

Reviewer 1

The work by Martin et al described the isolation of benzophenone-3 (BP3) degrading strain Rhodococcus sp. USK10 and the identification by 16S rRNA and ANI analysis. This is a nice paper in the topic. Here are some minor comments.

  • A chemical structure of BP3 was recommended to be shown in Fig 1, since not all readers realize what benzophenone-3 is.

We have added the chemical structure of BP3 in figure 1.

  • Two plasmids should be shown. They were long enough.

We agree with the reviewer that the 2 plasmids are long enough to merit graphical presentation. We have created 2 maps, which we have added to the text. Our main concern was that one of the plasmids is linear and CGviewer does not accommodate for linear constructs.

  • The public database accession No.s should be shown in order to make readers to repeat them.

Accession numbers to have been added in the “Data availability statement”.

  • Some references were cited incorrectly. For example, line 229-231, original refs for two Sphingomonas strains should be cited.

Original references have now been cited.

  • Line 261, do they have names of two plasmids? Give the name and corresponding accession numbers.

Sphingomonas wittichii DC-6 has 11 plasmids and the genes responsible for degradation of aromatic compounds are spread in genetic loci with mobile genetic elements. We would advise readers to follow up on the cited references for their specific applications.

  • References should be in the same style. For example, Journal names of ref 4 and 34 were in short form.

This was an artifact from the reference manager we used. We have corrected these mistakes along a couple more that we found.

  • Is a CONCLUSION section needed?

A conclusion is not absolutely necessary. Following the reviewers’ remarks we decided to remove the last two sentences.

Reviewer 2 Report

General comment

This is an interesting study that examined the BP3-biodegradation potential of Rhodococcus sp. USK10. While the results are interesting and relevant for environmental bioremediation, some aspects (e.g. BP3 biodegradation) were not discussed sufficiently in the context of available literature. See specific comments below.

Results and Discussion

Lines 137-153: The discussion is very shallow. The relevance of this study to biotechnological application of USK10 for bioremediation of BP3 (and by extension other organic pollutants) should be sufficiently discussed in the context of available literature. In addition, the potential benefits of bioaugmentation using bacterial isolates and consortium in comparison to natural attenuation of organic pollutants may also be added. In this regard, below are relevant and recent literature that can be used:

https://doi.org/10.1016/j.envint.2008.07.018; https://doi.org/10.3390/genes12010098; https://doi.org/10.1016/j.chemosphere.2021.133143 and https://doi.org/10.1080/15320383.2016.1190313.

Line 149: Remove the hyphen in “demon-strates”.

Line 156: Remove the hyphen in “repli-cates”.

Finally, I suggest that the biodegradation portion of the paper (currently section 3.1) be placed last (i.e. after the discussion of genome analysis/annotation and phylogenetic placement). In that way, the presentation would flow better. It will also make the “results and discussion” to harmonize with the word order of the title.

Author Response

We thank the two reviewers for their valuable input. We have gone meticulously through the each comment, and corrected the manuscript accordingly. Please see the detailed replies to each comment below:

Reviewer 2

General comment

This is an interesting study that examined the BP3-biodegradation potential of Rhodococcus sp. USK10. While the results are interesting and relevant for environmental bioremediation, some aspects (e.g. BP3 biodegradation) were not discussed sufficiently in the context of available literature. See specific comments below.

We have taken the reviewer’s comments into consideration and we have expanded our discussion in section 3.1.

Results and Discussion

  • Lines 137-153: The discussion is very shallow. The relevance of this study to biotechnological application of USK10 for bioremediation of BP3 (and by extension other organic pollutants) should be sufficiently discussed in the context of available literature. In addition, the potential benefits of bioaugmentation using bacterial isolates and consortium in comparison to natural attenuation of organic pollutants may also be added. In this regard, below are relevant and recent literature that can be used:

https://doi.org/10.1016/j.envint.2008.07.018; https://doi.org/10.3390/genes12010098; https://doi.org/10.1016/j.chemosphere.2021.133143 and https://doi.org/10.1080/15320383.2016.1190313.

Following are answer above, we have expanded the section and we have cited some of the suggested literature. We thank the reviewer for providing valuable feedback.

  • Line 149: Remove the hyphen in “demon-strates”.

Corrected.

  • Line 156: Remove the hyphen in “repli-cates”.

Corrected.

  • Finally, I suggest that the biodegradation portion of the paper (currently section 3.1) be placed last (i.e. after the discussion of genome analysis/annotation and phylogenetic placement). In that way, the presentation would flow better. It will also make the “results and discussion” to harmonize with the word order of the title.

We can definitely see the point of the reviewer. However, we would prefer not to rearrange the order of the manuscript as the starting point and probably most interesting part is the fact that Rhodococcus USK10 can in fact degrade BP-3. This was our starting point to provide a complete genome sequence.

Round 2

Reviewer 2 Report

The revision is adequate.